# Automatic Configuration of Benchmark Sets for Classical Planning

**Álvaro Torralba,**[1] **Jendrik Seipp,**[2] **Silvan Sievers**[2]

[1]Aalborg University, Denmark
[2]University of Basel, Switzerland
alto@cs.aau.dk, jendrik.seipp@unibas.ch, silvan.sievers@unibas.ch

## Abstract

The benchmarks from previous International Planning Competitions are commonly used to evaluate new planning algorithms. Since this set has grown organically over the years, it has several flaws: it contains duplicate tasks, unsolvable tasks, trivially solvable domains, and domains with modelling errors. Also, diverse domain sizes complicate aggregating results. Most importantly, however, the range of task difficulty is very small in many domains. We propose an automated method for creating benchmarks that solves these issues. To find a good scaling in difficulty, we automatically configure the parameters of benchmark domains. We show that the resulting benchmark set improves empirical comparisons by allowing to differentiate between planners more easily.

## 1 Introduction

Domain-independent planning aims to develop general solvers that find solutions to arbitrary sequential decision-making problems. This makes the evaluation of planners an essential part of planning research. The International Planning Competition (IPC) has set some evaluation standards and triggered the development of tools that compare planners in terms of different metrics (Linares López, Celorrio, and Helmert 2013; Seipp et al. 2017; Vallati, Chrpa, and McCluskey 2018). The most popular metric is coverage, i.e., the number of solved benchmark instances within certain time and memory limits. Typically, there are two main goals for the evaluation: (1) analyze the impact of the novel algorithms by comparing their performance against a baseline, and (2) compare the performance against the state of the art to evaluate the progress in the area.

Evaluating planners on different benchmark sets may produce different results, leading to different conclusions from the evaluation. Not only is it important which domains we choose, but also how we model the domains (Riddle, Holte, and Barley 2011), and which instances of a domain we select. Therefore, having a standardized benchmark set is important to increase the comparability of results across different papers, and to avoid the use of benchmarks tailored for the proposed technique.

We focus on classical planning where the current standard benchmark set has grown across the nine editions of the IPC so far, from 1998 to 2018 (e.g., Hoffmann and Edelkamp 2005; Linares López, Celorrio, and Olaya 2015). Numerous researchers have contributed to this set by carefully designing new domains (e.g., Hoffmann et al. 2006), so it features a diverse set of domains that pose interesting challenges for planning algorithms. However, there are several issues with this benchmark set (Moraru and Edelkamp 2019). For example, it uses a different number of instances per domain, which reduces the value of statistics aggregated over different domains. Moreover, instances in the current benchmark set were scaled to be useful for the evaluation of planners at the respective IPC. Some of the domains are trivially solved by modern planners, making it impossible to show any coverage advantages over a baseline. On the other hand, early IPC editions did not have a specialized track for optimal planning, and some of their instances are too hard even for state-of-the-art optimal planners.

This paper deals with the question of how to generate instances of a domain to evaluate planning algorithms. Our goal is to improve the empirical evaluation of future planning papers by (1) providing an algorithm for automatically constructing interesting benchmark sets and by (2) using this algorithm to construct a new benchmark set where differences in performance are better reflected in coverage than under the current standard. We aim to generate a set of instances that range from very easy (solved by most planners) to very hard (out of reach for current state-of-the-art planners) allowing future approaches to show benefits with respect to the harder instances. This definition necessarily depends on the algorithms being evaluated.

We identify which properties are desirable for a benchmark set and propose an automatic method that generates a set of instances, given an *instance generator*, a *baseline* planner that represents the expected minimum performance of any planner, and a set of *state-of-the-art* planners. The instance sets generated by our method fulfill the desirable properties by design. To avoid overfitting to the sets of planners used and not introduce a bias in our benchmark set, our method does not select a set of instances directly, but rather performs a search on the space of possible parameters for the generator to obtain a set of instances of adequate difficulty. We use our tool to design two separate sets of benchmarks, for optimal and satisficing planning, and show their advantages over the current standard IPC benchmark set.

## 2 Background

Informally, a classical *planning task* is defined by an initial state, a set of actions and a goal description. Given a planning task, a *planner* finds a *plan*, that is, a sequence of actions that can be applied in the initial state to achieve the goal. A plan is optimal if it minimizes the summed-up cost of the actions among all plans. If the planner is guaranteed to find an optimal solution, it is an optimal planner, otherwise it is a satisficing planner. In both settings, we only consider solvable planning tasks.

Since its inception in 1998, the International Planning Competition (IPC) has set the standards for the evaluation of planners such as the planner input language PDDL (McDermott et al. 1998). The IPC also introduced numerous planning tasks from different problem settings, called *domains*.

A planning task is typically divided into a domain and an instance file. The domain file defines the types of objects, their properties, and the action schemas. Each instance file can have a different number of objects, initial state and goals. Most domains have an instance generator, a program that, given certain parameters and a random seed, will generate a new instance of the domain. Even though many instance generators are available,[1] most planning papers use the benchmarks introduced for the IPCs, since a standardized benchmark set makes research more reproducible.

As an example, consider the Nomystery domain, where a truck must deliver a set of packages to certain locations. To do that, there is a limited amount of fuel that is consumed by drive actions. Instances differ in the amount of fuel available, the number of locations and their connections, the number of packages, and their initial and final location. The instance generator for Nomystery accepts several parameters that allow the benchmark designer to control the difficulty of the generated instances: the number of locations, the number of packages, the number of edges between locations, the maximum fuel consumption between two locations, and the constrainedness $C \geq 1$, so that the amount of fuel in the initial state is set to $C$ times the minimum fuel consumption required to solve the instance.

## 3 Benchmark Design Principles

The purpose of a benchmark set is to evaluate planners and compare their performance on a diverse class of problems. Ideally, one should select a diverse set of domains that are representative of real-world scenarios where different users apply planning to solve their problems. However, in addition to selecting interesting domains, one must select a set of concrete instances from each domain to evaluate the planners on. This selection of instances is an important step in the design of the benchmark set, since different instance sets of a domain may lead to different conclusions about which planner is better at solving instances of a given domain. Our goal is that, for any two planners $A$ and $B$ (possibly unknown at the time when the instance set is generated) if $A$ is consistently faster than $B$ on the instances of a domain, the probability that this is reflected on the coverage score should be as high as possible.

---

[1]https://github.com/AI-Planning/pddl-generators

|  | IPC | | | New'14 | | |
|---|---|---|---|---|---|---|
|  | L | D | O | L | D | O |
| Nomystery | 11 | **20** | 12 | 25 | **30** | 24 |
| Rovers | **40** | **40** | **40** | 22 | 18 | 21 |
| Woodworking | **50** | **50** | **50** | 18 | 27 | **30** |
| Total | 101 | **110** | 102 | 65 | **75** | **75** |

Table 1: Coverage of LAMA (L), and two IPC 2018 agile planners Decstar (D) and OLCFF (O) on three domains.

For aggregated statistics to be meaningful, not only should all domains have the same number of instances, but their difficulty should also scale similarly. Otherwise, conclusions taken from the empirical evaluation may be biased. Table 1 shows an example comparing 3 planners in 3 domains when using the IPC instances and our New'14 benchmark set, as described in the evaluation section. A paper evaluating these planners with IPC instances would reach the conclusion that Decstar is clearly superior to the other two planners in these domains, both in total coverage and on a per-domain basis since it has better or equal coverage in all domains. However, this conclusion is biased because instances are not well scaled. Instances in Rovers and Woodworking are way too easy and therefore they do not show any differences between the planners. Using our New'14 instances leads to a different conclusion: all three planners are complementary. Of course, no strong conclusions can be taken out of only 3 domains. However, using more domains will help to alleviate this issue only if the instances are well scaled.

A good scaling must meet three conditions: (1) have easy instances that are solved by all planners, (2) have hard instances that are not solved by any current planner, and (3) the instance difficulty should grow smoothly.

Condition (1) is necessary for experiments to be informative at all: if some planners do not solve any instance, no conclusions can be obtained about their relative performance. This happens in some domains of the IPC benchmark set for optimal planning. E.g., Fišer, Torralba, and Shleyfman (2019) write that "In childsnack, [they] measured about twice as many expanded states per second. However, no planner solved any instance in this domain.".

Condition (2) is necessary for new algorithms to show that they can deal with instances that previous planners could not, as shown by our example in Table 1.

Condition (3) is necessary for differences between the planners' performance to be reflected in coverage. To see why, consider an idealized setting where a baseline planner $A$, whose runtime scales exponentially ($t(A, x) = x^C$ for some constant $C$), is compared to an improved planner version $B$ which is always faster than $A$ by at least a factor of $K > 1$, i.e., $t(B, x) \leq \frac{t(A,x)}{K}$. Given these assumptions, there is a guaranteed difference in coverage if and only if (1) some instances are solved by $B$, (2) not all instances are solved by $A$, and (3) $K \geq C$. Otherwise, there may be cases where both planners solve the same number of instances,

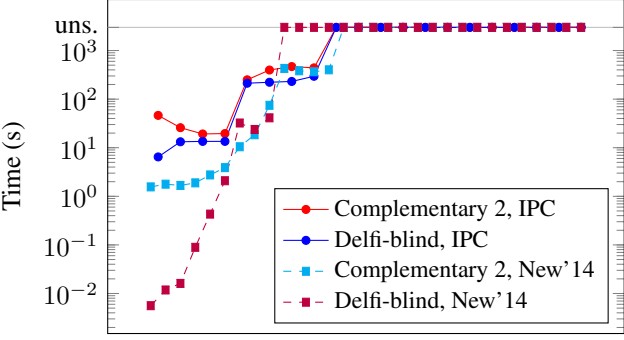

Figure 1: Runtime of two IPC 2018 optimal planners in the Barman domain using the IPC and New'14 instance sets.

and the difference in performance by a factor of $K$ is missed by the coverage analysis. If these conditions do not hold, it is possible to choose runtimes for $A$ and $B$ that are compatible with these exponential scalings, so that their coverage is equal. For example, if $K = 2$ and $C = 3$, then (3) does not hold. So for any time limit (e.g., 300 seconds), if the runtime of the last instance solved by $A$ is close enough to the time limit (e.g., 250 seconds), the next instance cannot be solved by $B$ below the time limit (e.g., $\frac{250 \cdot 3}{2} > 300$).

Real distributions of planner runtimes over sets of instances differ from this idealized example in that they usually involve constant factors, the runtime scaling of different planners may be completely different, and even for a single planner it may be impossible to obtain instances that scale according to the desired runtimes in some domains. But ideally, all domains should consist of a collection of instances of increasing difficulty, ranging from very easy to very hard for current planners. Therefore, we aim for a collection where the easiest instance is quickly solved by most planners; all domains have instances that are not solved by current planners; and difficulty scales by approximately a factor of 1.5–2 between consecutive instances.

Figure 1 exemplifies why a smooth scaling is important in practice. The plot shows the runtimes of two optimal planners on the Barman domain from IPC 2011 and our New'14 benchmark set. In the IPC instances the difficulty does not grow smoothly. Instead, for each group of four instances the difficulty increases visibly and the runtime of all planners increases by about one order of magnitude. This is undesirable since we cannot observe differences in performance for some planners by inspecting their coverage. In contrast, the difficulty on the new benchmark set grows more smoothly, there are instances of more varied difficulty for all planners, and fewer jumps in their runtime. Accordingly, now we can observe that Complementary 2 is able to solve some instances that are not solved with Delfi-blind in this domain.

Given our definition of an ideal benchmark set as one that meets conditions (1), (2), and (3) described above, the instance selection necessarily depends on the planning algorithms being evaluated. As our objective is to generate a benchmark set to evaluate future planners that do not exist yet, one cannot directly select instances that are useful to

compare planner's right now. However, selecting an instance set that scales well for current planners may generalize well for planners that are introduced in the next few years. Indeed, our New'14 instance set featured in our examples of Table 1 and Figure 1 was configured without using any planner after 2014, so it did not use any information regarding the IPC'18 planners mentioned in our examples.

One must be careful not to "overfit" the benchmark set to match the set of selected planners, so that the difficulty scales well for the selected planners but not for future planners. To avoid overfitting, we impose two restrictions on the benchmark configuration process. On the one hand, the optimization process does not consider concrete instances, but rather only decides which overall characteristics they should have (e.g. the number of objects in each instance). The final benchmark set is then generated with a random seed different from the one used during our optimization process. On the other hand, we do not consider the individual results of all planners available for the optimization. In each domain, we require a baseline planner that represents the expected minimum performance of any planner to ensure that some instances are solved by all planners. Also, to ensure that some instances remain unsolved, we estimate the performance of state-of-the-art planners by taking the minimum runtime of any of the available planners on each instance. This makes our instance selection as objective as possible since it does not depend on the concrete set of planners available to the benchmark designer, as long as the best planner for the given domain is considered.

## 4  Configuration of Planning Benchmarks

We consider domains that have an instance generator with several parameters to control the hardness of the generated instances.

### 4.1  Framework

We model the problem of generating the instances of a benchmark set as follows. Our tool takes as input a tuple $(spec, G, \mathcal{A}, \mathcal{B})$, where $spec$ is a domain specification; $G$ is an instance generator; $\mathcal{A}$ is a set of state-of-the-art planners; and $\mathcal{B}$ is a set of baseline planners. The output will be a set of instances of that domain.

The domain specification describes the instance generator parameters and their constraints and it is discussed in detail in the next section. The instance generator $G$ is a function that takes as input a tuple of parameter values $\rho = \langle \rho_1, \ldots, \rho_k \rangle$ and a random seed $seed \in \mathbb{N}^+$ and outputs a planning task. Let $p \in \mathcal{A} \cup \mathcal{B}$ be a planner, and $\Pi$ a planning task produced by $G(\rho, seed)$ for some $seed \in \mathbb{N}^+$. We define $t(p, \Pi)$ as the runtime of planner $p$ to solve task $\Pi$. For every instance we characterize the performance of a set of planners $\mathcal{A}$ on an instance $\Pi$ as the minimum runtime of any planner, $t(\mathcal{A}, \Pi) = \min_{p \in \mathcal{A}} t(p, \Pi)$.

Given a parameter configuration $\rho$, we define $t(p, \rho)$ as the average runtime over all possible tasks that the generator may output for different random seeds. We estimate $t(p, \rho)$ by sampling $k$ tasks $\Pi_1, \ldots, \Pi_k$ from the distribution and taking the average running time $\frac{\sum_{i \in [1,k]} t(p, \Pi_i)}{k}$. In practice,

a small $k$ is sufficient for most domains. In our experiments we used $k = 1$, which was enough to produce robust results.

Our goal is to select a set of parameter configurations $\rho^1, \ldots, \rho^n$ such that the running time $t(\mathcal{A}, \rho^i)$ scales smoothly with $i$, as described in our general design principles. Note that our automatic tool is not allowed to hand-pick the random seed, but rather the final benchmark set is created by sampling these distributions of instances with new random seeds. This helps to avoid overfitting. An assumption is that the variance of the runtimes $t(\mathcal{A}, \Pi_i)$ for tasks generated with the same parameter configuration is not too high, since otherwise the parameters provided to the generator are irrelevant for obtaining a smooth difficulty scaling. This is a reasonable assumption in practice, following analyses made in the context of predicting planner runtimes (de la Rosa, Cenamor, and Fernández 2017). Out of 10 domains analyzed by de la Rosa, Cenamor, and Fernández, most of them had a very low variance. The one with highest variance was Barman, where 90% of the instances were still relatively close to the average runtime, especially for our purposes.

## 4.2 Domain Specification

In order to use our benchmark configuration tool, the benchmark designer must specify how to call the instance generator, what parameters are available, as well as which values are appropiate for each parameter.

We distinguish between two types of parameters. *Linear parameters* can be assigned arbitrary non-negative numeric values, where larger values usually result in harder instances. They are typically used to specify the number of objects of a given type. Each generator should have at least one linear parameter that helps to control the difficulty of the generated instances. In contrast, *enumerated parameters* have a finite set of values, and we do not make any assumption about their impact on instance hardness. All other parameters are fixed to a predefined constant value.

We define the instances of a domain as a set of *sequences* of instances. A sequence consists of a list of planning tasks $\Pi_1, \Pi_2, \ldots$ of increasing difficulty. To ensure that difficulty increases, all instances in the sequence have a fixed value for all enumerated parameters, whereas the value of linear parameters increases linearly across the sequence. We specify this via the base value $b$ that the linear parameter takes for $\Pi_1$ and the slope $m$. For example, suppose that a domain has two linear parameters that define the number of packages ($b = 2, m = 1$), and trucks ($b = 1, m = 0.5$). Then, the sequence will generate instances with the following numbers of packages and trucks: $(2, 1), (3, 1), (4, 2), (5, 2), (6, 3)$, etc.

Considering sequences of instances allows us to choose the parameters that generate instances which current planners fail to solve within reasonable time. A limitation of this approach is that not all combinations of parameters are possible. In the example above, a single sequence cannot contain both $(3, 2)$ and $(2, 3)$ because that would require to decrease one of the parameters, which is not allowed by our linear scaling. In most cases, this is not a problem because we can use multiple sequences of instances for a single domain. A notable exception are parameters that define the

```
generator_command = "nomystery -l {locations}
  -p {packages} -n {edgefactor} -m {edgeweight}
  -c {constrainedness} -s {seed} -e 0"
domain_attributes = [
  LinearAttr("locations", lower_b=3, upper_b=5,
                          lower_m=0.1, upper_m=1),
  LinearAttr("packages", lower_b=2, upper_b=10),
  ConstantAttr("edgefactor", "1.5"),
  ConstantAttr("edgeweight", "25"),
  EnumAttr("constrainedness", [1.1, 1.5, 2.0])]
```

Figure 2: Example of a domain specification with the generator command and the specification of the corresponding parameters.

width and height of a grid, because they have a strong interaction, i.e., the number of cells is the product of both parameters. In that case, we consider them a single parameter so that the number of tiles in the grid scales linearly.

The snippet from Figure 2 shows the domain specification for the nomystery domain. For each linear parameter, lower and upper bounds for the base and slope values should be provided. This allows the domain modeller to specify preferences on which parameters to scale (e.g., by restricting the slope $m$ for the number of locations to be between 0.1 and 1, they indicate their preference to increase difficulty by scaling the number of packages). Note that this is important, since a property of a good benchmark set is that instances reflect problems that are "interesting in practice", and this is a subjective matter that the configuration tool cannot decide on its own. If the benchmark designer has no such preference, all parameters can be left with a default interval.

Often, instance generators impose constraints on the range of parameter values or their combination. Those constraints must be enforced by adding a postprocessing function that updates the value of the parameters passed to the generator. This is an arbitrary function provided by the benchmark designer which receives the parameters that were automatically chosen and outputs the final parameters that will be provided to the generator. For example, if the number of packages has to be greater than the number of locations, instead of directly selecting the number of packages, our linear scaling will consider the number of locations and the number of additional packages. All of these adjustments must be done on a per-domain basis, since they depend on the specific characteristics of the domain and generator.

Given this framework, our automatic tool decides which sequences of instances are suitable for each domain. This is done in two phases: the first phase designs a set of candidate sequences (Sequence Optimization), and the second phase performs a final selection that adheres to our design principles as much as possible (Sequence Selection).

## 4.3 Sequence Optimization

The first phase generates sequences of 30 instances by optimizing sequence parameters. To guide the search towards sequences that scale the instance difficulty as smoothly as possible, we compute a penalty score for each sequence and

search for the sequence that minimizes this score.

Sequences are evaluated by running hand-picked state-of-the-art ($\mathcal{A}$) and baseline ($\mathcal{B}$) planners on their instances, using a time limit of 180 seconds per instance. We ignore instances that are solved under 10 seconds, considering that differences of $\pm 5$ seconds are not meaningful enough. Since the sequences are generated with increasing values of the linear parameters, we assume that the runtimes will always increase, so we can stop our evaluation as soon as one instance is not solved under the time limit. In cases where this does not hold, we enforce it by sorting the runtimes of the instances. Our assumption is that these anomalies stem from using different random seeds for the instance generator and the results will be different with different random seeds.[2] The runtime of a set of planners is the minimum of the runtimes of the individual planners. For evaluating a sequence we consider the first five instances with a minimum runtime above 10 seconds. We ignore harder instances because they will usually incur runtimes above the 180 seconds time limit. Let $t(X, 1), \ldots, t(X, 5)$ be the runtimes of the set of planners $X$ on the first five instances with a runtime above 10 seconds. The penalty score is defined as $\sum_{i \in [2,5]} S(\mathcal{B}, i) + S(\mathcal{A}, i)$ where $S(X, i) =$

$$
\begin{cases}
3 - \dfrac{2t(X, i)}{t(X, i-1)} & \text{if } 1 \le \dfrac{t(X, i)}{t(X, i-1)} \le 1.5 \\[2ex]
0 & \text{if } 1.5 < \dfrac{t(X, i)}{t(X, i-1)} \le 2 \\[2ex]
1 - \dfrac{2t(X, i-1)}{t(X, i)} & \text{if } 2t(X, i-1) \le t(X, i) \le 180 \\[2ex]
2 & \text{if } t(X, i) > 180
\end{cases}
$$

This penalty is lower for sequences whose runtime scales smoothly, assigning a minimum score of 0 to any sequence where the runtimes of both the baseline and state-of-the-art planners scale exponentially with a factor between 1.5 and 2, e.g., $\langle 10, 15, 23, 35, 52, \ldots \rangle$, or $\langle 10, 20, 40, 80, 160, \ldots \rangle$. If not enough instances are solved in the $[10, 180]$ second interval, the sequence gets a penalty of 2, and otherwise we assign it a penalty between 0 and 1. To avoid generating sequences where all instances are solved by the state-of-the-art planners, we also add a penalty of 1 for each instance solved by them beyond 20 instances. To guarantee that all valid sequences contain some instances solvable within the time limit and to speed up the evaluation we require the first three instances to be solved within 10, 60, and 180 seconds, respectively. Otherwise, we discard the sequence, unless all linear parameters are at their minimum value.

The concrete choice of penalty values is arbitrary. What matters is that sequences that minimize this score adhere more to the design principles introduced in Section 3 than those that do not, thereby guiding the parameter optimization towards good sequences.

---

[2]Note that any parameter that has an unpredictable influence on the runtime of a planner should be considered an enumerated parameter and remain constant for a given sequence.

## 4.4 Sequence Selection

After performing one or more optimization runs for a domain (using different random seeds) as described above, we collect all sequences seen during the optimization process. Since this set can be very large, we only keep the 100 sequences with the lowest penalty score per value of the enumerated parameters. For each group of sequences where the planners solves the same instances, we only keep one member of the group. This filtering ensures that we keep a set of diverse sequences with a good penalty score.

For each sequence, we collect the runtimes of all instances solvable in 180 seconds from the sequence optimization phase. For the rest of the instances, we estimate the runtime by assuming that runtimes will increase according to the average increasing factor $t(\mathcal{A}, i)/t(\mathcal{A}, i-1)$ observed on the instances solved between 5 and 180 seconds. This is a very rough estimate but it is accurate enough for the purposes of choosing up to when a sequence should be continued (see below).

We model the problem of selecting a suitable set of sub-sequences as a Mixed-Integer Programming (MIP) problem, where constraints directly aim to model the design principles of Section 3. The decision variables model the start and end points of each sub-sequence of instances. The selection must satisfy the following *hard constraints* that model properties desirable for a good set of instances:

(H1) The number of selected instances must be exactly 30.

(H2) There must be at least one instance solvable by the baseline under 30 seconds.

(H3) All sequences must start with an instance that is solvable by a state-of-the-art planner and end with an instance whose estimated runtime is higher than 2000 seconds.

(H4) Each parameter configuration must be used (with different random seeds) at most twice, and only once for domains whose generators do not admit a random seed.

The objective is to minimize the summed-up penalty score of all sequences used, plus the penalty incurred for violating any of the following *soft constraints*:

(S1) The number of instances solved by the baseline under 30 seconds must be between 2 and 6 (with a penalty of $2x^2$ where $x$ is the deviation with respect to the constraint).

(S2) The number of instances solved under 180 seconds must be between 8 and 15 (with a penalty of $2x^2$ where $x$ is the deviation with respect to the constraint).

(S3) All sequences must end with an instance whose estimated runtime is between 18 000 and 180 000 seconds (that is, 1–2 orders of magnitude larger than the typical time limit of 30 minutes). Larger times $t$ incur a penalty of $100t/180000$ and smaller times incur a penalty of $100(18000/t)$.

(S4) If a parameter configuration is used more than once, there is a penalty of 100.

| | Optimal | Satisficing |
|---|---|---|
| Configuration New'14 | blind search (baseline), all four components of the FDSS 1 portfolio from IPC 2011 (Helmert et al. 2011) and SymBA$_1^*$ from IPC 2014 (Torralba et al. 2014) | greedy best-first search with FF heuristic (baseline, Hoffmann and Nebel 2001), LAMA (Richter and Westphal 2010) and Madagascar (Rintanen 2012) |
| Configuration New'20 | union of *Configuration New'14* and *Evaluation* | union of *Configuration New'14* and *Evaluation* |
| Evaluation | five components of Delfi1 portfolio from IPC 2018 using symmetry pruning and partial order reduction (blind search, iPDB, LM-Cut and two M&S variants, see Katz et al. 2018) and three vanilla IPC 2018 planners: Complementary2 (Franco, Lelis, and Barley 2018), DecStar (Gnad, Shleyfman, and Hoffmann 2018), Scorpion (Seipp 2018b) | eight vanilla IPC 2018 planners: Cerberus (Katz 2018), BFWS-PREF, DUAL-BFWS and POLY-BFWS (Francès et al. 2018), DecStar (Gnad, Shleyfman, and Hoffmann 2018), OLCFF (Fickert and Hoffmann 2018), Fast Downward Remix (Seipp 2018a) and Saarplan (Fickert et al. 2018) |

Table 2: Choice of planners for benchmark generation and evaluation.

For domains where all instances in the sequence are solved by state of the art planners under 180 seconds (because the domain is solvable in polynomial time and it is impossible to fulfill our criteria with state-of-the-art planners), we consider the runtimes of the baseline instead of those of the state of the art planners in our constraints described above.

Constraints (H2), (S1) and (S2) ensure that the instance set contains some easy instances, so that any future planning algorithms are expected to solve at least some instances, allowing researchers to analyze the behaviour of their algorithms in the domain. Constraints (H3) and (S3) ensure that, whenever possible, at least some of the instances are expected to be out of reach for state-of-the-art planners. Together with minimizing the penalty score of the selected sequences, they aim to obtain a smooth scaling, since sequences must interpolate between easy and hard instances and sequences with smoother scaling are preferred. Finally, constraints (H4) and (S4) are needed to avoid duplicate instances and instances that are very similar to each other.

The penalties are set arbitrarily, but they scale quadratically with respect to the deviation because it is better to not fulfill several soft constraints entirely than to completely ignore one of the constraints.

## 5   Experiments

We implemented the first phase, i.e., sequence optimization, using the automatic configurator SMAC (Hutter, Hoos, and Leyton-Brown 2011). We test our approach by running two completely separated optimizations for optimal and satisificing planners. As baseline planners, we use blind search for optimal planning and greedy best-first search with the FF heuristic (Hoffmann and Nebel 2001) for satisficing planning, both implemented in Fast Downward (Helmert 2006a).

Both for optimal and satisficing planning, we generate two separate benchmark sets, New'14 and New'20, that differ in the set of state-of-the-art planners available for optimization. The New'14 set consists of a heterogeneous set of planning algorithms from IPCs 2011 and 2014. The New'20 set is trained using the same planners plus the ones used in the evaluation. Therefore, for New'14, the configuration and evaluation sets are disjoint, while for New'20, the sets overlap. Table 2 gives an overview of the planners used.

Since we limit each planner run during the optimization to 3 minutes, we adapt the planners for the configuration phase by breaking portfolios into components and by adapting preprocessing time limits. For optimization in each domain, we hand-pick 1–3 planners that perform best in that domain, which is sufficient to approximate the minimum time of any planner in each instance. We run SMAC 10 times using different random seeds. Each run is limited to 10 hours. After the first phase finishes, we consider all sequences encountered during optimization for the second phase, i.e., sequence selection. We filter the instances as described in Section 4.4 and solve the MIP for sequence selection using CPLEX 12.10, which finishes in under 30 seconds for each domain.

We evaluate the new benchmark sets using the aforementioned planners, limiting each run to 30 minutes and 3.5 GiB. In Table 3 we compare the IPC benchmarks to the new benchmark sets for optimal and satisficing planning. We compare benchmark sets according to two metrics: the range of coverage scores per domain, which allows us to see how many instances are solved by all planners and how many remain unsolved by any of the planners; and the number of pairwise comparisons in which a planner had higher coverage than another, which quantifies how many differences in the performance of planners are reflected by the coverage score.

In optimal planning the difference between the benchmark sets is rather subtle because difficulty typically scales very fast with increasing instance size. Therefore, the IPC set has some interesting instances in all domains. Also, it can be very hard to generate instance sequences whose difficulty scales smoothly, since often increasing one of the parameters of a generator by a unit has a big impact on runtime.

The results are more pronounced for satisficing planners, where the IPC set scales very poorly for some domains. Only in the Elevators domain the IPC set is superior in terms of comparisons detected by the coverage score compared to both new benchmark sets. In contrast, with the new benchmark sets, we observe differences in performance in domains like Blocksworld, Driverlog or Zenotravel, where all planners solve all instances in the IPC set. Overall, New'14 uncovers more differences in coverage between pairs of planners than the IPC set in 21 out of 26 domains, while

| Optimal | #IPC | coverage range | | | comparisons | | | Satisficing | #IPC | coverage range | | | comparisons | | |
|---|---|---|---|---|---|---|---|---|---|---|---|---|---|---|---|
| | | IPC | '14 | '20 | IPC | '14 | '20 | | | IPC | '14 | '20 | IPC | '14 | '20 |
| barman | 34 | 4–11 | 9–13 | 9–12 | 12 | **21** | 19 | barman | 40 | 39–40 | 7–25 | 9–30 | 7 | 24 | **27** |
| blocksworld | 35 | 18–30 | 5–12 | 5–12 | 18 | **24** | 24 | blocksworld | 35 | 35–35 | 7–24 | 4–22 | 0 | 27 | **28** |
| childsnack | 20 | 0–6 | 9–20 | 6–21 | 12 | 18 | **22** | childsnack | 20 | 1–20 | 14–30 | 2–19 | 27 | 25 | **28** |
| data-network | 20 | 6–14 | 5–12 | 5–16 | **27** | 25 | 27 | data-network | 20 | 9–19 | 10–30 | 13–30 | 24 | **27** | 25 |
| depot | 22 | 5–14 | 9–25 | 8–16 | **26** | 26 | 24 | depot | 22 | 21–22 | 12–20 | 11–26 | 7 | **27** | 22 |
| driverlog | 20 | 7–15 | 6–30 | 5–18 | 22 | **26** | 25 | driverlog | 20 | 20–20 | 29–30 | 9–19 | 0 | 12 | **24** |
| elevators | 50 | 28–44 | 7–14 | 10–18 | **26** | 26 | 23 | elevators | 50 | 49–50 | 30–30 | 30–30 | **7** | 0 | 0 |
| floortile | 40 | 16–34 | 9–18 | 8–17 | 21 | 21 | **22** | floortile | 40 | 4–40 | 1–12 | 1–11 | 17 | **25** | 24 |
| grid | 5 | 1–3 | 6–26 | 4–21 | 19 | **28** | 27 | grid | 5 | 5–5 | 4–20 | 9–21 | 0 | **26** | 24 |
| gripper | 20 | 8–20 | 11–30 | 11–30 | **7** | **7** | **7** | gripper | 20 | 20–20 | 26–30 | 26–30 | 0 | **7** | **7** |
| hiking | 20 | 12–18 | 7–9 | 5–16 | 23 | 15 | **25** | hiking | 20 | 10–20 | 2–22 | 3–26 | 24 | **28** | 27 |
| logistics | 63 | 13–34 | 5–17 | 5–14 | **27** | 27 | 25 | logistics | 63 | 51–63 | 5–30 | 5–26 | 17 | **27** | 26 |
| miconic | 150 | 56–142 | 4–28 | 3–30 | 25 | 27 | **28** | miconic | 150 | 150–150 | 30–30 | 30–30 | **0** | **0** | **0** |
| nomystery | 20 | 8–20 | 3–27 | 5–21 | 18 | **28** | 27 | nomystery | 20 | 12–20 | 19–30 | 2–30 | 23 | 18 | **26** |
| openstacks | 130 | 42–71 | 4–11 | 3–7 | **24** | 18 | 7 | openstacks | 160 | 99–160 | 12–21 | 14–23 | 21 | **27** | 25 |
| parking | 40 | 0–15 | 11–18 | 12–21 | **28** | 24 | 23 | parking | 40 | 36–40 | 14–20 | 13–16 | 7 | **24** | 21 |
| rovers | 40 | 6–13 | 4–26 | 6–19 | **25** | 22 | 7 | rovers | 40 | 38–40 | 10–22 | 6–30 | 7 | 26 | **27** |
| satellite | 36 | 7–14 | 8–30 | 4–27 | 22 | 25 | **26** | satellite | 36 | 26–36 | 5–30 | 6–14 | **23** | 17 | **23** |
| scanalyzer | 50 | 21–33 | 6–16 | 7–15 | **27** | 24 | 24 | scanalyzer | 50 | 48–50 | 9–16 | 13–14 | 12 | **21** | 12 |
| snake | 20 | 7–14 | 5–20 | 7–19 | 22 | **24** | 21 | snake | 20 | 3–17 | 6–30 | 5–14 | 27 | **28** | 26 |
| storage | 30 | 15–18 | 9–25 | 2–19 | 21 | **27** | 26 | storage | 30 | 21–30 | 6–26 | 7–17 | 26 | **27** | 26 |
| tpp | 30 | 7–20 | 7–30 | 2–7 | **24** | **24** | 21 | tpp | 30 | 29–30 | 10–26 | 6–21 | 15 | **27** | 27 |
| transport | 70 | 24–35 | 5–30 | 8–19 | 21 | 18 | **22** | transport | 70 | 65–70 | 22–30 | 15–23 | 7 | 24 | **26** |
| visitall | 40 | 12–30 | 6–21 | 5–20 | **27** | **27** | **27** | visitall | 40 | 36–40 | 4–30 | 4–29 | 7 | 24 | **26** |
| woodworking | 50 | 38–50 | 16–25 | 10–14 | 22 | **26** | 24 | woodworking | 50 | 28–50 | 6–30 | 5–30 | 13 | **27** | 27 |
| zenotravel | 20 | 8–13 | 6–30 | 3–13 | 23 | 26 | **28** | zenotravel | 20 | 20–20 | 6–29 | 5–17 | 0 | 23 | **25** |

Table 3: Comparison of the IPC and new benchmark sets for optimal and satisficing planning. The #IPC column shows the number of tasks per domain in the IPC benchmark set, which is always 30 for the new sets. The coverage range shows the minimum and maximum coverage of any planner. In the "comparisons" columns we list how many pairs of planners yield different coverage scores for each benchmark set.

the opposite is the case in only 4 domains.

The comparison between New'14 and New'20 reveals that our technique is not very sensitive to the set of state-of-the-art planners. The reason is that the state of the art has not advanced enough in the last six years to make a set of instances trained with our method in 2014 outdated.

## 6 Discussion

Our paper deals with the problem of generating instances that are adequate to evaluate planning algorithms. The goal is to select instances that scale well, so that differences in algorithm performance are reflected in the number of problems solved within a certain time limit. It must be remarked that no benchmark set can replace a careful analysis of the results. Aggregating results from different domains without further analysis may be misleading and an empirical analysis only based on total coverage should be discouraged. Nevertheless, coverage is a useful metric to summarize experimental data and it is used by most planning papers. As shown by our experiments, the coverage metric is more meaningful for the benchmark sets generated with our approach than with the previous standard. Our main result is a new benchmark set, as well as a set of generators and tools that can be used

in the future to automatically generate new instances.

In other communities like SAT, there has been a lot of research on how to construct random instances (Selman, Mitchell, and Levesque 1996; Achlioptas et al. 2000; Giráldez-Cru and Levy 2015; Xu et al. 2005) around the phase transition (Cheeseman, Kanefsky, and Taylor 1991). Our approach is orthogonal to any approach that can generate new instances, e.g., around the phase transition of planning problems (Rintanen 2004; Rieffel et al. 2014), or with suitable initial states and goals for Sokoban (Bento, Pereira, and Lelis 2019). Those approaches provide an instance generator that adjusts the instance difficulty for a given problem size, but to generate an instance set still requires to select the value of certain parameters. Our approach is complementary, since it can be used to select suitable values that are useful to evaluate a given set of solvers. Our tool can also be adapted to generate benchmarks with different characteristics, e.g., with smaller instances that are solved in a few seconds (Ruml 2010). Future work could also consider relations among different domains theoretically (Helmert 2003, 2006b) or empirically (Cenamor and Pozanco 2019).

## Acknowledgments

We thank Florian Pommerening for helping us set up the experiments and we thank the anonymous reviewers for their helpful comments. We have received funding for this work from the European Research Council (ERC) under the European Union's Horizon 2020 research and innovation programme (grant agreement no. 817639). Álvaro Torralba was employed by Saarland University and the CISPA Helmholtz Center for Information Security during part of the development of this paper.

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
