# OpenReview forum: "Automatic Configuration of Benchmark Sets for Classical Planning"
_icaps-conference.org/ICAPS/2020/Workshop/HSDIP — HSDIP 2020_

### Official Review · AnonReviewer2 · 2020-03-31
**Early assessment**

**Rating:** 8
**Confidence:** 4

**Review:**

This paper addresses the automated generation of problem instances. The aim is to obtain a benchmark set consisting of instances that make an empirical differentiation of different planners possible. For this purpose, the problem of creating a set of instances is modeled as an optimization problem.

The topic of the paper fits the workshop, since one of the typical topics of the HSDIP workshop is the investigation of “challenging domains for existing combinations of heuristics and search algorithms”.

Overall, I like the paper and I believe that an automated procedure to obtain a benchmark set that is well suited to measure different planners is an interesting and totally relevant topic. While reading the paper, some remarks and questions came to my mind:

1) The sentence “Evaluating planners on different benchmark sets may produce different results, leading to different conclusions about the evaluation”  reminded me of a work by Riddle et al. (2011) which showed that the ranking of the International Planning Competition could be influenced if the representation of a single domain (Blocksworld) is different.

2) Some machine learning techniques require a large amount of data for training. I would be interested to know if this automated technique also makes it possible and is well suited to create a large amount of instances for a domain, i.e. not just 30, but hundreds or thousands?

3) In the conclusion, it is noted that the proposed approach is orthogonal to any approach that can generate new instances. I am not sure if this means that the approach is not compatible with the construction of random instances? For example, if we consider different models for the construction of random instances proposed by Rintanen (2004), it seems that such models also have several parameters that could be optimized in a similar way as the IPC domains.

Minor comments:
- Introduction: aggregated over*y* different
- Sequence Optimization: in Section _ than those
- Sequence Selection: principles of Section _.

References:
- P. J. Riddle, R. C. Holte, M. W. Barley: Does Representation Matter in the Planning Competition? SARA 2011
- J. Rintanen: Phase Transitions in Classical Planning: An Experimental Study. ICAPS 2004: 101-110

---

> ### Author Response · Authors · 2020-07-23
> **Thanks for the review**
>
> Thanks for the review and useful comments.
>
> 1) Yes, we have added the reference since how the domains are modeled is indeed one of
> the factors that can influence the conclusions of the evaluation.
>
> 2) Automatically generating instances for learning is a very interesting question, but our
> techniques do not apply there directly. Of course one can use the same generators we use
> for these purposes. But our focus is on how to generate sets of instances to evaluate
> planners, which has different requirements for the generated instances. For generating
> training data one does not need sequences of instances of increasing difficulty; and one
> does not need to consider instances out of reach for current planners. One can just
> randomly select values for the instance generator and, if the task is too hard, reduce the
> values until finding instances solvable in a reasonable amount of time.
>
> 3) On the contrary. With being orthogonal we meant that both are complementary: our
> technique could indeed help to select suitable parameters for generators like that one. We
> have rephrased it to make it more clear, and incorporated Rintanen's reference.

---

> > ### Comment · AnonReviewer2 · 2020-08-17
> > **Final Remarks**
> >
> > Thanks for answering my questions and comments and for considering them. I still like the paper and I think it fits the workshop very well.
> >
> > While reading the new revision of the paper, I only found a few minor comments.
> >
> > - In constrast => In contrast (page 3)
> > - ... only dedide which ... => only decide which (page 3)
> > - On page 4 there is a large space before the sentence "Also, any constraint imposed by the generator..." begins.
> > - The enumeration on page 5 ((H1) - (S4)) exceeds the column margin.

---

### Official Review · AnonReviewer1 · 2020-08-17
**Review (revised paper version)**

**Rating:** 3
**Confidence:** 3

**Review:**

This paper proposes a scheme for selecting sequences of parameters for
planning problem instance generation, within a specific domain, aimed
at creating a set of instances that better measure the scaling
performance of planning systems.

I see some problems with this proposal, or the way it is presented in
the paper. First, it does not make any consideration for the
variability in instance hardness with equal parameters, which is a
known phenomenon in many domains. TSP-like domains, like Logistics,
Satellite and Rovers, for example often exhibit this characteristic.
The hardness of these problems depends as much on the specific
arrangement of objects in the initial state, roadmap connectivity,
location of goals, etc, as on the number of such, so that the
variation in hardness of different instances that have the same "size"
can be much greater than that resulting from increasing size. The
authors suggest treating the random seed, which implicitly determines
the specific arrangement of an instance for given size parameters, as
an enumerated parameter, and thus keeping it fixed during parameter
sequence generation. However, they also state that the final instance
set "will be generated with a different random seed than the one used
during our optimization process", which implies that there may be no
relation between the hardness of a parameter setting observed during
sequence construction and that of the corresponding instance in the
final set.

The assumption that increasing linear parameters results in harder
problems, even "typically", is also somewhat problematic, since it
ignores relationships between parameters and "phase transition"-like
effects. For example, consider a simple domain that involves
navigating a maze with obstacles, and a parameter that determines the
density of obstacles. At low densities, problems can be easy because
relaxations that ignore obstacles are accurate; at high densities,
problems are easy to solve, or prove unsolvable, because the reachable
state space is very small. It is somewhere in the middle of the range
that problems are the hardest. Note that "density", in this example,
could be defined implicitly, for instance with separate parameters for
maze size and number of obstacles. While this is a made-up example,
examples that have a similar behaviour appear in some IPC domains, for
example the number of instruments in the Satellite domain.

Second, the reliance on a selection of "state of the art" planners in
shaping the benchmark set can also be problematic, because scaling in
different parameters may affect different planners in different ways.
An example of this can be seen by comparing the behaviour of planners
on the IPC 1998 and 2000 Logistics instance sets. The IPC 2000 set
scales up problems mainly by increasing the number of goal (packages);
the number of cities and airplanes grows very little (from 3 to 5, and
1 to 2, respectively). The IPC 1998 set, on the other hand, scales up
the number of all types of objects in the instances. Experimental
comparisons that separate these two instance sets sometimes have shown
a difference in how planners perform on them. If the reference set
includes only planners that, for example, scale the same with
increasing numbers of objects of any type, the resulting benchmark set
will not be able to exhibit the performance of a planner that scales
much better in the number of goals while other parameters are limited
to small values. Another example can be found in the results of the
2016 unsolvability competition, which showed that some planners
although not performing well on average across the set of domains used
in the competition, did very well, and better than the overall
top-performers, in certain domains.

I do not think there is a single best way to select a set of benchmark
problems, except perhaps if you first formulate the hypothesis under
test and select according to that, but in that case the planners or
algorithms would be known, the instances could be tailored to their
capability, and the choice of domains as important as the choice of
instances. Selecting an instance set "blindly" is perhaps a problem
for organisers of the IPC, but even then it does not need to be
suitable for more than the current state of the art, and we know not
to over-interpret the results. If the authors want to propose a method
for selecting benchmark problems for experiments, they should first
clearly delineate the purpose of the experiment that the selected
benchmarks are to be used for - what is the scientific question that
it will answer? - and secondly demonstrate the potential failure modes
of the selection method against that purpose.

---

> ### Author Response · Authors · 2020-08-23
> **Thanks for your feedback**
>
>     Thank you for the feedback! We disagree in your overall evaluation. As we detail
>     below, our method aims at answering a specific scientific question: how to select
>     instances of a domain to create a benchmark set that can be used to evaluate planning
>     algorithms in the next few years, assuming that the evaluation of those papers will be
>     conducted as in previous years (i.e., coverage based evaluation under the IPC time and
>     memory limits)?
>
>     This is an important problem that has a very direct practical application: improve the
>     evaluation of planning algorithms in the near future.
>
>     You make some good points regarding specific considerations that we'll try to clarify in this response and/or in the final version of the paper.
>
> I do not think there is a single best way to select a set of benchmark problems, except
> perhaps if you first formulate the hypothesis under test and select according to that, but
> in that case the planners or algorithms would be known, the instances could be tailored to
> their capability, and the choice of domains as important as the choice of
> instances. Selecting an instance set "blindly" is perhaps a problem for organisers of the
> IPC, but even then it does not need to be suitable for more than the current state of the
> art, and we know not to over-interpret the results. If the authors want to propose a
> method for selecting benchmark problems for experiments, they should first clearly
> delineate the purpose of the experiment that the selected benchmarks are to be used for -
> what is the scientific question that it will answer? - and secondly demonstrate the
> potential failure modes of the selection method against that purpose.
>
>     We already describe the purpose of the experiments in the paper. We have
>     specific considerations in mind, described in the "design principles" section,
>     and our experiments aim to evaluate whether the new benchmark set satisfies
>     those principles.
>
>     We assume that the domains have been designed beforehand, by taking the previous IPC
>     domains and their generators. We assume the typical evaluation of planning algorithms,
>     where two or more algorithms are run on a standard set of benchmarks and compared in terms
>     of total coverage to determine which algorithm is superior (for such a benchmark set) and
>     discover interesting trends. We also assume that the coverage of multiple domains will get
>     aggregated in a total coverage to argue that one algorithm is superior in practice. Note
>     that there are hundreds of planning papers that follow these assumptions, so they are not
>     unrealistic.
>
>     In such a setting, we argue that generating instances in a similar way for all domains is
>     important to make total coverage meaningful. If each domain has a completely different
>     amount of instances, total coverage becomes meaningless. Not only that: if each domain has
>     been balanced differently, e.g., by introducing a lot of easy or hard instances, total
>     coverage may also become meaningless. Having a method that automatically selects instances
>     following the same criteria for all domains is useful so that all domains are balanced in
>     a similar way.
>
>     The specific question is, for a given domain, do the differences in planner performance
>     get reflected in their coverage?  We tailor our method to create a benchmark set that is
>     arguably superior according to this criteria to the IPC one that everyone uses.
>
>     But at the same time, we also consider that maximizing this metric is not the only
>     important consideration, and try to avoid introducing a huge bias, e.g., due to the planners
>     selected for our optimization process, or manually selecting instances/random seeds. We
>     try to measure this bias experimentally too, by creating two different instance sets with
>     all planners and with only a subset and comparing the results.
>
>
>
> I see some problems with this proposal, or the way it is presented in the paper. First, it
> does not make any consideration for the variability in instance hardness with equal
> parameters, which is a known phenomenon in many domains. TSP-like domains, like Logistics,
> Satellite and Rovers, for example often exhibit this characteristic. The hardness of these
> problems depends as much on the specific arrangement of objects in the initial state,
> roadmap connectivity, location of goals, etc, as on the number of such, so that the
> variation in hardness of different instances that have the same "size" can be much greater
> than that resulting from increasing size. The authors suggest treating the random seed,
> which implicitly determines the specific arrangement of an instance for given size
> parameters, as an enumerated parameter, and thus keeping it fixed during parameter
> sequence generation. However, they also state that the final instance set "will be
> generated with a different random seed than the one used during our optimization process",
> which implies that there may be no relation between the hardness of a parameter setting
> observed during sequence construction and that of the corresponding instance in the final
> set.
>
>     We will incorporate a brief discussion about this in the paper. In short, we do not
>     assume that all instances with the same parameters have exactly the same runtime, and
>     as you point out this may cause instances with more objects to be easier than smaller
>     instances. But what we expect is that, on average, larger instances will be harder.
>
>     We do not treat the random seed as an enumerated parameter. Instead, our tool uses a
>     different random seed in every call to the generator. Our tool also has an option to
>     generate multiple instances with the same parameter configuration and take the average
>     runtime out of those.
>
>     Our aim is not to create the "perfect" benchmark set for the current set of planners.
>     That could introduce too much bias, e.g., towards selecting random seeds that are
>     suitable for the planners we're using. Instead, we just try to ensure that the
>     instances scale "reasonably well", so it does make sense to accept some noise in the
>     distribution of instance hardness for a given set of parameters.
>
>
> The assumption that increasing linear parameters results in harder problems, even
> "typically", is also somewhat problematic, since it ignores relationships between
> parameters and "phase transition"-like effects. For example, consider a simple domain that
> involves navigating a maze with obstacles, and a parameter that determines the density of
> obstacles. At low densities, problems can be easy because relaxations that ignore
> obstacles are accurate; at high densities, problems are easy to solve, or prove
> unsolvable, because the reachable state space is very small. It is somewhere in the middle
> of the range that problems are the hardest. Note that "density", in this example, could be
> defined implicitly, for instance with separate parameters for maze size and number of
> obstacles. While this is a made-up example, examples that have a similar behaviour appear
> in some IPC domains, for example the number of instruments in the Satellite domain.
>
>     Yes, the tool cannot identify these relations among the parameters, and they must be
>     spotted by the domain designer. In your example, there should only be a linear
>     parameter that declares the size of the maze and an enumerated parameter,
>     density. Note that if two linear parameters are used, there is still the need of
>     setting "number of obstacles < maze size" to avoid an error in the instance generator,
>     so one way or another the domain modeler needs to take the relationship between
>     these two parameters into account.
>
>
> Second, the reliance on a selection of "state of the art" planners in shaping the
> benchmark set can also be problematic, because scaling in different parameters may affect
> different planners in different ways. An example of this can be seen by comparing the
> behaviour of planners on the IPC 1998 and 2000 Logistics instance sets. The IPC 2000 set
> scales up problems mainly by increasing the number of goal (packages); the number of
> cities and airplanes grows very little (from 3 to 5, and 1 to 2, respectively). The IPC
> 1998 set, on the other hand, scales up the number of all types of objects in the
> instances. Experimental comparisons that separate these two instance sets sometimes have
> shown a difference in how planners perform on them. If the reference set includes only
> planners that, for example, scale the same with increasing numbers of objects of any type,
> the resulting benchmark set will not be able to exhibit the performance of a planner that
> scales much better in the number of goals while other parameters are limited to small
> values. Another example can be found in the results of the 2016 unsolvability competition,
> which showed that some planners although not performing well on average across the set of
> domains used in the competition, did very well, and better than the overall
> top-performers, in certain domains.
>
>     Our tool will select the scaling that is smoother in difficulty, according to the best
>     planner for every instance. This only introduces an unfair bias if the best planner
>     for a given scaling of a domain has not been considered during the
>     optimization. That's why incorporated as many planners as possible in our evaluation,
>     instead of only considering the IPC winners.  For us, a "state of the art" planner is
>     any planner that achieves the best results in a number of instances of some domain,
>     even if it performs poorly in others. We'll clarify this in the paper.
>
>     When setting the parameters of a domain, one can restrict the scaling to a subset of
>     parameters, or even force that some parameters should always be scaled. We consider
>     that part of the domain definition. Creating a new benchmark set for everyone is not
>     an easy task, since it is full of decisions like those. We tried to follow similar
>     scalings as the ones used by the IPC organizers. However, note that the decisions
>     made by the IPC organizers were often arbitrary and should not be glorified as the
>     ground truth. That's why this is something we would like to discuss in HSDIP before
>     releasing our new benchmark set.

---

### Comment · Program_Chairs · 2020-09-14
**Final Decision: Accept**

Dear Authors,

Thank you very much for your submission. We are happy to inform you that we have decided to accept it and we look forward to your talk in the workshop. You will receive additional information per mail in the coming days.

Best,
The HSDIP'20 team

---

### Decision · Program_Chairs · 2020-09-30

Accept